# Study on Bioactive Components of Aromatic *Cynanchum thesioides* (Freyn) K. Schum by Solvent Fractionation

**DOI:** 10.3390/plants13223123

**Published:** 2024-11-06

**Authors:** Tao Lyu, Woonjung Kim

**Affiliations:** Department of Chemistry, University of Hannam, Daejeon 34430, Republic of Korea; lyutao@naver.com

**Keywords:** aromatic *Cynanchum thesioides*, antioxidant activity, DPPH radical scavenging activity, ABTS radical scavenging activity, solvent fraction, yield

## Abstract

This study evaluated the antioxidant activity of the methanol extract from aromatic *Cynanchum thesioides* (Freyn) K. Schum. by fractionating it with different solvents, aiming to provide theoretical evidence for the development of products related to aromatic *Cynanchum thesioides* (Freyn) K. Schum. The yield of the methanol extract was 13.33%, with the water fraction showing the highest yield, followed by n-hexane, n-butanol, dichloromethane, and ethyl acetate. Among these fractions, the ethyl acetate fraction showed the highest antioxidant activity, as indicated by total polyphenol content and ABTS radical scavenging activity. The DPPH radical scavenging activity, reducing power, and SOD-like activity measurements showed no significant difference between the ethyl acetate and n-butanol fractions, as both showed the highest radical scavenging activity. In the FRAP activity measurement, the n-butanol fraction ranked behind the ethyl acetate and dichloromethane fractions in terms of antioxidant activity. Although the ethyl acetate fraction showed the highest antioxidant activity, its yield was only 1.29%, making it unsuitable for product production considering productivity and economy. However, the n-butanol fraction showed overall high antioxidant activity and was approximately four times more abundant, with a yield of 5.80% compared to the ethyl acetate fraction. Consequently, considering both productivity and economy, the n-butanol fraction is considered suitable for product development and production.

## 1. Introduction

Reactive oxygen species (ROS) generated in the body can damage cells and tissues, leading to the inactivation of proteins, attacks on unsaturated fatty acids, and overall deterioration of bodily functions. These processes contribute to oxidation, ageing, and the development of cardiovascular and neurological diseases [1]. Consequently, there is a demand for the development of antioxidants that can either enhance the body’s antioxidant defense system or regulate ROS [2]. However, synthetic antioxidants such as butylated hydroxyanisole (BHA) and butylated hydroxytoluene (BHT) have been found to have toxic and carcinogenic side effects, limiting their use. Therefore, there is a need for safer natural antioxidants with fewer side effects [3]. Aromatic medicinal herbs, among various natural plants, have traditionally been used for disease prevention and treatment. In recent years, research on the antioxidant activities of aromatic medicinal herbs has been ongoing. For example, *Cynanchum auriculatum*, *Cynanchum bungei*, and *Cynanchum wilfordii*, which are native to East Asia and belong to the Asclepiadaceae family, have been used as traditional aromatic medicinal herbs and functional dietary supplements for hundreds of years. Studies have demonstrated their excellent antioxidant activity, immune-boosting properties, and anticancer effects [3,4].

*Cynanchum thesioides* (Freyn) K. Schum (hereinafter referred to as *Cynanchum thesioides*), also a member of the Asclepiadaceae family, is a perennial herb that stands erect or climbs, reaching heights of 15 to 25 cm. It contains white latex, and its stems are slender and weak, often branching extensively [5]. *Cynanchum thesioides* grows on hillsides, dunes, wastelands, and farmland, with its flowering period from June to September and fruiting period from August to October. It is distributed across northern China, Mongolia, and Siberia [6]. In several regions of Inner Mongolia and Mongolia, the fruits of *Cynanchum thesioides* are consumed as food. The whole plant is traditionally used in folk medicine to promote lactation, reduce fever, and alleviate inflammation and pain. It has been a key ingredient in remedies for abdominal pain and diarrhea for centuries [7]. Additionally, *Cynanchum thesioides* has applications as food, fodder, and industrial raw material, and it is valuable for maintaining water and soil in pastures, holding significant economic and ecological importance [8].

Previous studies have highlighted *Cynanchum thesioides*’s bioactive compounds and pharmacological effects, such as triterpenoids, flavonols, steroids, phenolic acids, amyrin, and oleanolic acid, which contribute to its antioxidant and anti-inflammatory properties [9,10]. Notably, quercetin, a representative flavonol, is known for its antioxidant and anti-ageing effects, and thesioide oside, a unique compound of *Cynanchum thesioides*, has been reported to possess various pharmacological effects, including cell protection and potential neuroprotective benefits [11]. While several studies have focused on the plant’s phytochemistry and traditional uses, recent research has also explored its therapeutic potential. For example, studies have demonstrated that extracts of *Cynanchum thesioides* possess significant anti-inflammatory and antioxidative properties, suggesting its utility in pharmaceutical applications [12]. However, studies evaluating and comparing the antioxidant activities of its solvent fractions are limited.

This study aims to assess the yield and antioxidant activities of the methanol extract of *Cynanchum thesioides* and its sequential n-hexane, dichloromethane, ethyl acetate, n-butanol, and water fractions. By examining these fractions, this research provides insights into the antioxidant efficacy of various bioactive components, contributing to a theoretical basis for the development of food, pharmaceutical, and cosmetic products using *Cynanchum thesioides* as a natural aromatic medicinal herb resource.

## 2. Results and Discussion

### 2.1. Yield of Methanol Extract and Solvent Fractions

In the measurement of antioxidant activity, the yield of natural aromatic plant extracts holds significant importance in terms of productivity and cost-effectiveness, as the extraction of antioxidant components can vary depending on the solubility differences of the solvents used. To evaluate the components of the methanol extract from *Cynanchum thesioides*, sequential fractionation was performed using organic solvents with different polarities, namely n-hexane, dichloromethane, ethyl acetate, n-butanol, and water. The yield of the methanol extract from *Cynanchum thesioides* was found to be 13.33%. The yields of the fractions obtained with each organic solvent (on a dry basis) are presented in Table 1. The water fraction showed the highest yield at 13.92%, followed by n-hexane, n-butanol, dichloromethane, and ethyl acetate fractions with yields of 7.25%, 5.80%, 2.15%, and 1.29%, respectively.

### 2.2. Total Polyphenol Content

Polyphenols are aromatic compounds with two or more phenolic hydroxyl (-OH) groups in one molecule, including flavonoids, tannins, catechins, and others. These compounds give color to aromatic plants and possess functional activities such as antioxidant and anticancer properties [13]. The total polyphenol content of each fraction from *Cynanchum thesioides* is shown in Table 2. The results indicate that the ethyl acetate fraction had the highest polyphenol content, at 112.54 ± 0.59 mg gallic acid equivalents (GAE)/g (*p* < 0.05), followed by n-butanol, dichloromethane, methanol, n-hexane, and water fractions with contents of 100.86 ± 0.63, 61.05 ± 0.78, 37.37 ± 0.20, 17.01 ± 0.20, and 5.92 ± 0.30 mg GAE/g, respectively (*p* < 0.05). This suggests that the phenolic compounds in the methanol extract of *Cynanchum thesioides* are predominantly present in the ethyl acetate fraction.

According to the study by Wu et al., the ethyl acetate fraction of *Cynanchum bungei* Decne., a medicinal aromatic plant in the Asclepiadaceae family, showed the highest total polyphenol content, at 42.38 mg/g [14]. This indicates that phenolic compounds such as quercetin, which are abundantly present in aromatic plants of the Asclepiadaceae family, interact well with ethyl acetate, making them easily soluble. Consequently, many studies have used ethyl acetate as a solvent for extraction and fractionation [15,16]. Given that antioxidant activity is often correlated with total polyphenol content, the ethyl acetate fraction’s rich variety of phenolic compounds, including quercetin, tamarixetin, and thesioide oside, is likely to contribute significantly to the observed antioxidant activity. Thus, our findings imply that the ethyl acetate fraction not only contains high levels of total polyphenols but also plays a crucial role in enhancing the antioxidant potential of *Cynanchum thesioides*.

### 2.3. Ferric Reducing Antioxidant Power (FRAP)

The ferric reducing antioxidant power (FRAP) measures the antioxidant activity of a sample based on the reduction of the Fe^3+^-TPTZ complex to Fe^2+^-TPTZ at low pH, which reflects the degree of hydroxylation and polyphenol binding [17]. The FRAP measurement results for each fraction of *Cynanchum thesioides* are shown in Table 3. The ethyl acetate fraction exhibited the highest FRAP value, at 3.49 ± 0.00 M/g (*p* < 0.05), followed by dichloromethane, n-butanol, methanol, n-hexane, and water fractions, which showed reducing powers of 2.32 ± 0.01, 2.24 ± 0.01, 1.52 ± 0.01, 1.20 ± 0.00, and 0.93 ± 0.00 M/g, respectively (*p* < 0.05). Li reported a positive correlation between the antioxidant activity of phenolic substances and FRAP values; the trend observed in this study’s FRAP results is consistent with the total polyphenol content results for *Cynanchum thesioides* [18]. Therefore, it is inferred that the highest FRAP value of the ethyl acetate fraction in this study is due to its highest total polyphenol content.

### 2.4. DPPH Radical Scavenging Activity

The DPPH (2,2-Diphenyl-1-picrylhydrazyl) radical scavenging activity assay is widely used to evaluate antioxidant activity. It measures the reduction and decolorization of the stable free radical DPPH by various compounds, making it a convenient method for assessing the activity of antioxidants such as ascorbic acid, cysteine, glutathione, tocopherol, polyhydroxy aromatics, and aromatic amines [19]. The DPPH radical scavenging activity of the *Cynanchum thesioides* methanol extract and its solvent fractions is expressed as IC_50_ values in Table 4. The ethyl acetate and n-butanol fractions exhibited the lowest IC_50_ values, 0.42 ± 0.00 mg/mL and 0.48 ± 0.01 mg/mL, respectively, indicating the highest antioxidant activities among the methanol extract and its fractions (*p* < 0.05). Following these were the dichloromethane, methanol, water, and n-hexane fractions, with IC_50_ values of 0.73 ± 0.00, 1.22 ± 0.00, 2.24 ± 0.03, and 6.12 ± 0.41 mg/mL, respectively (*p* < 0.05). The IC_50_ value for the positive control, ascorbic acid, was 0.01 ± 0.00 mg/mL.

The results indicate that the ethyl acetate fraction, which had the highest total polyphenol content, also exhibited the highest DPPH radical scavenging activity. This is consistent with reports by Lee [20] and Kim [21] that higher polyphenol and flavonoid content in extracts correlates with increased antioxidant activity. The strong correlation between the elevated total phenol content and DPPH scavenging activity suggests that phenolic compounds play a critical role in enhancing the antioxidant potential of the ethyl acetate and n-butanol fractions. This finding aligns with the understanding that flavonols and triterpenoids, known for their antioxidant properties, are abundantly dissolved in these fractions of *Cynanchum thesioides*. Therefore, the methanol extract of *Cynanchum thesioides* exhibited the highest DPPH radical scavenging activity when fractionated using ethyl acetate or n-butanol.

### 2.5. ABTS Radical Scavenging Activity

The ABTS radical scavenging activity assay is based on the generation of the activated cation radical ABTS+ through the reaction of ABTS with peroxidase and H_2_O_2_. The ABTS+ radical, which is blue green, is reduced and decolored by receiving electrons from antioxidant substances. This decolorization is then used to evaluate the antioxidant activity [22]. The ABTS radical scavenging activity of the *Cynanchum thesioides* methanol extract and its solvent fractions is expressed as IC_50_ values in Table 5. The ethyl acetate fraction showed the lowest IC_50_ value of 0.43 ± 0.01 mg/mL, indicating the highest scavenging activity (*p* < 0.05). This was followed by the n-butanol, dichloromethane, methanol, water, and n-hexane fractions, with IC_50_ values of 0.60 ± 0.00, 0.92 ± 0.01, 1.61 ± 0.00, 3.48 ± 0.01, and 9.45 ± 0.02 mg/mL, respectively (*p* < 0.05). The IC_50_ value for positive control, ascorbic acid, was 0.07 ± 0.00 mg/mL.

Consistent with the results of the DPPH radical scavenging activity assay, the ethyl acetate fraction, which had the highest total polyphenol content, also exhibited the highest ABTS radical scavenging activity. This supports the findings of Kainama [23] and Sanchez [24], who reported a strong positive correlation between high antioxidant activity and high total polyphenol content in ethyl acetate fractions. The observed relationship between total phenol content and ABTS scavenging activity further emphasizes the significant contribution of phenolic compounds to the antioxidant efficacy of the ethyl acetate fraction.

Interestingly, the water fraction also demonstrated increased ABTS activity, which could be attributed to the presence of hydrophilic antioxidant compounds that effectively scavenge ABTS radicals. This finding raises questions about the specific compounds responsible for this activity in the water fraction. To further elucidate the mechanisms behind these observations, we plan to incorporate analytical methods such as HPLC in our future experiments to identify and quantify the antioxidant compounds in both the water and ethyl acetate fractions. This will enhance our understanding of the antioxidant properties of *Cynanchum thesioides* and the role of its polyphenolic composition in mediating these effects.

### 2.6. Reducing Power

Reducing power is an indication of the antioxidant capacity of a substance, where antioxidants stabilize free radicals by donating hydrogen to the ferric-ferricyanide (Fe^3+^) complex, reducing it to the ferrous (Fe^2+^) state. This reduction can be quantified by measuring absorbance; higher absorbance values correspond to stronger reducing power and result in a more greenish color [25]. The EC_50_ values of the reducing power of the *Cynanchum thesioides* methanol extract and its solvent fractions are presented in Table 6. The ethyl acetate and n-butanol fractions exhibited the most potent reducing power, with EC_50_ values of 0.87 ± 0.00 and 0.95 ± 0.00 mg/mL, respectively, among the methanol extract and its fractions (*p* < 0.05). Following these were the dichloromethane, methanol, water, and n-hexane fractions, with EC_50_ values of 1.38 ± 0.03, 1.40 ± 0.01, 1.92 ± 0.00, and 10.36 ± 0.30 mg/mL, respectively (*p* < 0.05). The EC_50_ value for positive control, ascorbic acid, was 0.04 ± 0.00 mg/mL. It is known that the reducing power activity of antioxidants is closely related to their ability to inhibit the formation of peroxides, prevent the binding of transition metals, and scavenge radicals [26]. This aligns with the results of this study, which showed similar trends in the reducing power and DPPH and ABTS radical scavenging activities. Therefore, it was confirmed that higher DPPH and ABTS radical scavenging activities correspond to higher reducing power in the *Cynanchum thesioides* methanol extract and its fractions.

### 2.7. Superoxide Dismutase (SOD)-Like Activity

Superoxide dismutase (SOD) is an antioxidant enzyme that catalyzes the conversion of harmful reactive oxygen species within cells into hydrogen peroxide (H_2_O_2_) and oxygen (O_2_) through the reaction (2O_2_ + 2H → H_2_O_2_ + O_2_). The H_2_O_2_ generated by SOD is further converted into harmless water and oxygen molecules by peroxidase or catalase, thus protecting the body [27]. The results of SOD-like activity measurements for the fractions of *Cynanchum thesioides* are shown in Table 7. The ethyl acetate fraction exhibited the highest activity, at 30.36 ± 0.44% (*p* < 0.05), followed by the n-butanol, dichloromethane, water, n-hexane, and methanol fractions, with activities of 28.98 ± 1.52, 28.17 ± 1.12, 19.20 ± 0.93, 10.36 ± 0.30, and 14.72 ± 1.36%, respectively (*p* < 0.05). This study found a similar trend between the total polyphenol content and SOD-like activity measurements of the *Cynanchum thesioides* fractions. Previous reports by Kim [1] and Azuma [28] have indicated that higher polyphenol content is associated with greater SOD-like activity. It is speculated that this is due to the presence of quercetin and similar phenolic antioxidant compounds in the methanol extract and fractions of *Cynanchum thesioides*.

SOD is highly effective in eliminating free radicals and, therefore, has garnered significant interest in the pharmaceutical field for its superior efficacy compared to other antioxidants, and it is also widely used in anti-inflammatory and anti-ageing cosmetic products [1]. Consequently, the extract of *Cynanchum thesioides* could potentially aid in the removal of reactive oxygen species both in the body and in food products.

## 3. Materials and Methods

### 3.1. Materials and Extraction Preparation

*Cynanchum thesioides* was collected from Inner Mongolia, China and purchased air-dried, and the whole herb including stems, leaves and flower buds was cut into 10–15 mm lengths, mixed equally and used for the experiments and analyses. A sample of 30 g was subjected to three rounds of extraction at room temperature, each lasting 24 h, using methanol (99.5%, Samchun Pure Chemical Co., Ltd., Pyeongtaek, Republic of Korea) in a 10:1 (*w/v*) ratio. The extracts were filtered using filter paper (Whatman No. 4, Whatman International Ltd., Maidstone, UK). The filtrate was then subjected to methanol removal and concentration under reduced pressure using a rotary vacuum evaporator (EYELA SB-1000S, Rikakikai Co., Ltd., Tokyo, Japan) in a 40 °C heating bath. The methanol extract was dissolved in 200 mL of distilled water, as shown in Figure 1, and then mixed with n-hexane (95%, Samchun Pure Chemical Co., Ltd.) in a 1:1 (*v/v*) ratio. The mixture was fractionated using a separatory funnel and concentrated under reduced pressure with a rotary vacuum evaporator (EYELA SB-1000S, Rikakikai Co., Ltd.) in a 40 °C heating bath to obtain the n-hexane fraction. The same method was applied sequentially with dichloromethane (99.5%, Samchun Pure Chemical Co., Ltd.), ethyl acetate (99.5%, SK Chemicals Co., Ltd., Seongnam, Republic of Korea), n-butanol (99%, Samchun Pure Chemical Co., Ltd.), and water to obtain each fraction. All fractions were stored at temperatures below 4 °C and used in subsequent experiments. Each fraction was dissolved in DMSO (dimethyl sulfoxide, Samchun Pure Chemical Co., Ltd.) for the experiments.

### 3.2. Total Polyphenol Content

The total polyphenol content was measured by referencing the methods of Folin and Denis [29]. A mixture of Folin–Ciocalteu’s phenol reagent (Sigma-Aldrich Co., St. Louis, MO, USA) and distilled water at a ratio of 1:2 (*v/v*) was prepared. To this mixture, 40 μL of the sample was added and allowed to react in the dark at room temperature for 3 min. Subsequently, 600 μL of 10% Na_2_CO_3_ (Duksan Pure Chemical Co., Ltd., Ansan, Republic of Korea) was added, and the reaction was allowed to proceed in the dark at room temperature for 1 h. The absorbance was then measured at 765 nm using a xMark™ Microplate Absorbance Spectrophotometer (168-1150, Bio-Rad Laboratories, Hercules, CA, USA). The total polyphenol content was calculated by inserting the absorbance values into a standard curve prepared using gallic acid (Sigma-Aldrich Co.) as the standard. The results were expressed as mg of gallic acid equivalents (GAE) per gram of dry sample (mg GAE/g).

### 3.3. Ferric Reducing Antioxidant Power (FRAP)

The ferric reducing antioxidant power (FRAP) was measured based on the method of Benzie and Strain [30]. An acetate buffer (300 mM, pH 3.6) was prepared by mixing sodium acetate (Shimakyu’s Pure Chemical Co., Ltd., Osaka, Japan) and acetic acid (Samchun Pure Chemical Co., Ltd.). A 10 mM solution of 2,4,6-tris(2-pyridyl)-s-triazine (TPTZ, Sigma-Aldrich Co.) in 40 mM HCl (Samchun Pure Chemical Co., Ltd.) and a 20 mM solution of FeCl_3_·6H_2_O (Samchun Pure Chemical Co., Ltd.) were then mixed in a ratio of 10:1:1 (*v/v/v*). The mixture was incubated at 37 °C in an incubator (650D, Fisher Scientific, Ottawa, Canada) for 10 min. For the assay, 30 μL of the sample was mixed with 90 μL of distilled water and 0.9 mL of the FRAP reagent, and the reaction was carried out at 37 °C for 10 min. The absorbance was measured at 593 nm. The FRAP values were calculated using a standard curve prepared with FeSO_4_·7H_2_O (Samchun Pure Chemical Co., Ltd.), and the results were expressed as the molar concentration of FeSO_4_·7H_2_O per gram of sample.

### 3.4. DPPH Radical Scavenging Activity

The DPPH (2,2-Diphenyl-1-picrylhydrazyl) radical scavenging activity was measured based on the method of Blois [31]. The sample and a 0.2 mM DPPH solution (Sigma-Aldrich Co.) were mixed in a 1:1 (*v/v*) ratio and allowed to react in the dark at room temperature for 30 min. The absorbance was then measured at 517 nm. The DPPH radical scavenging activity was expressed as a percentage using the following Equation (1), and the IC_50_ value (the inhibitory concentration at which 50% of radicals are scavenged) was calculated and expressed in mg/mL. In this study, ascorbic acid (Samchun Pure Chemical Co., Ltd.), a known antioxidant, was used as a positive control for comparison.
DPPH radical scavenging activity (%) = [1 − (absorbance of sample)/(absorbance of control)] × 100(1)

### 3.5. ABTS Radical Scavenging Activity

The ABTS radical scavenging activity was measured based on the method of Fellegrini [32]. The ABTS reagent was prepared by adding 5 mL of distilled water to 88 μL of 140 mM K_2_S_2_O_8_ (Samchun Pure Chemical Co., Ltd.) and then adding two ABTS diammonium salt tablets (Sigma-Aldrich Co.). This mixture was allowed to react in the dark for 14–16 h. The resulting solution was then diluted with 95% ethanol (Samchun Pure Chemical Co., Ltd.) at a ratio of 1:88 (*v/v*) until the absorbance measured at 734 nm reached 0.7 ± 0.02. For the assay, 50 μL of the sample was mixed with 1 mL of the diluted ABTS reagent and allowed to react in the dark for 2 min and 30 s. The absorbance was then measured at 734 nm. The ABTS radical scavenging activity was expressed as a percentage using the following Equation (2), and the IC_50_ value (the inhibitory concentration at which 50% of radicals are scavenged) was calculated and expressed in mg/mL. In this study, ascorbic acid (Samchun Pure Chemical Co., Ltd.), a known antioxidant, was used as a positive control for comparison.
ABTS radical scavenging activity (%) = [1 − (absorbance of sample)/(absorbance of control)] × 100(2)

### 3.6. Reducing Power

The measurement of reducing power was conducted based on the method of Oyaizu [33]. To 125 μL of the sample, 125 μL of sodium phosphate buffer (200 mM, pH 6.6), prepared by mixing sodium phosphate monobasic (J.T. Baker Chemical Co., Phillipsburg, NJ, USA) and sodium phosphate dibasic (Sigma-Aldrich Co.), and 125 μL of 1% potassium ferricyanide (Samchun Pure Chemical Co., Ltd.) were sequentially added and mixed. The resulting mixture was then reacted in a 50 °C heating bath for 20 min, followed by cooling at room temperature for 10 min. After cooling, 125 μL of 10% trichloroacetic acid (Samchun Pure Chemical Co., Ltd.) was added, and the mixture was centrifuged (Smart R17, Hanil Science industrial Co., Ltd., Seoul, Republic of Korea) at 6000 rpm for 10 min. To 250 μL of the supernatant, 250 μL of distilled water and 25 μL of 1% ferric chloride (Samchun Pure Chemical Co., Ltd.) were added, and the absorbance was measured at 700 nm. The reducing power of the sample was expressed as a percentage (%) by converting the absorbance values to a range of 0 to 100%, where an absorbance range of 0 to 1 was considered equivalent to 0 to 100%. The EC_50_ (effective concentration) value, representing the sample concentration at which 50% reducing power is observed, was calculated and expressed in mg/mL.

### 3.7. Superoxide Dismutase (SOD)-Like Activity

The measurement of superoxide dismutase (SOD)-like activity was conducted based on the method of Marklund [34,35]. To 40 μL of the sample prepared at a concentration of 1 mg/mL, 520 μL of Tris-HCl buffer (pH 8.5), prepared by mixing Tris (50 mM, Sigma-Aldrich Co.) and EDTA (10 mM, Samchun Pure Chemical Co., Ltd.), and 40 μL of 7.2 mM pyrogallol (Sigma-Aldrich Co.) were thoroughly mixed. The mixture was then reacted at 25 °C for 10 min. Subsequently, 20 μL of 1 N HCl was added to stop the reaction, and the absorbance was measured at 420 nm. The results were expressed as a percentage (%), and the SOD-like activity was calculated using the following Equation (3):Superoxide dismutase (SOD)-like activity (%) = [1 − (A − B)/C] × 100(3)
where A = Absorbance at 420 nm determined with sample; B = Absorbance at 420 nm determined with buffer instead of pyrogallol; and C = Absorbance at 420 nm determined with buffer instead of sample.

## 4. Conclusions

In this study, we evaluated the antioxidant activity of different solvent fractions of *Cynanchum thesioides* methanol extract, aiming to provide a theoretical foundation for product development using this natural aromatic medicinal herb. The methanol extract yield was 13.33%, with the water fraction yielding the highest among the solvent fractions, followed by n-hexane, n-butanol, dichloromethane, and ethyl acetate. The ethyl acetate fraction exhibited the highest antioxidant activity, likely due to its elevated total phenol content, as indicated by ABTS radical scavenging activity measurements. Results for DPPH radical scavenging activity, reducing power, and SOD-like activity also showed no significant differences between the ethyl acetate and n-butanol fractions, both of which displayed the highest antioxidant activities, while the FRAP activity measurement indicated that the n-butanol fraction had the third-highest value.

However, despite the high antioxidant activity of the ethyl acetate fraction, its low yield of 1.29% limits its suitability for large-scale application due to practical and economic constraints. Conversely, the n-butanol fraction, with a significantly higher yield of 5.80% (more than four times that of ethyl acetate) and comparable antioxidant activity, offers a more practical and economically feasible option for commercialization. Based on these findings, we suggest that future research and product development focus on the n-butanol fraction, which provides an optimal balance of antioxidant efficacy and yield, supporting its potential as a viable candidate for further exploration in commercial applications.

## Figures and Tables

**Figure 1 plants-13-03123-f001:**
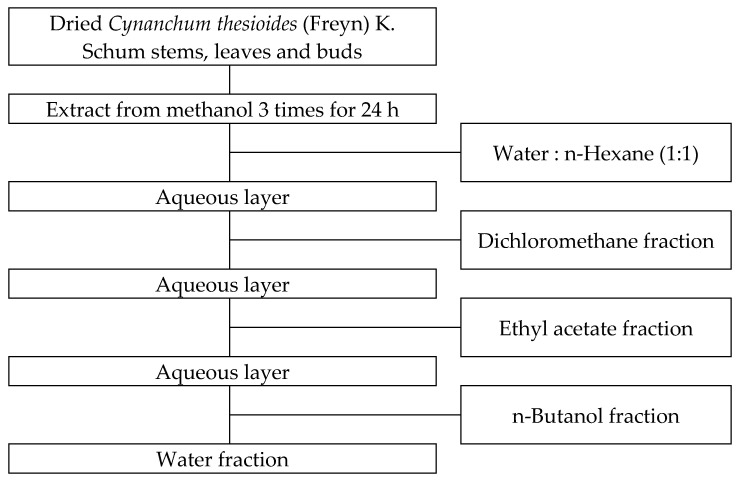
Procedure for fractionation of methanol extract from *Cynanchum thesioides* (Freyn) K. Schum stems, leaves and buds by various solvents.

**Table 1 plants-13-03123-t001:** The yields of solvent fractions extracted from *Cynanchum thesioides* (Freyn) K. Schum stems, leaves and buds.

Extraction Solvents	Yield (%, *w*/*w*)
Methanol ^1^	13.33
	**Fractions of methanol extract ^2^**
n-Hexane	7.25
Dichloromethane	2.15
Ethyl acetate	1.29
n-Butanol	5.80
Water	13.92

^1^ Yield (%) = weight of solid extract/weight of dry sample × 100; ^2^ Yield (%) = weight of solid fraction/weight of 99.5% methanol extract × 100.

**Table 2 plants-13-03123-t002:** Total polyphenol content of various solvent fractions from methanol extract of *Cynanchum thesioides* (Freyn) K. Schum stems, leaves and buds.

Extraction Solvent	Polyphenol Content (mg GAE ^1^/g)
Methanol	37.37 ± 0.20 ^2d3^
n-Hexane	17.01 ± 0.20 ^e^
Dichloromethane	61.05 ± 0.78 ^c^
Ethyl acetate	112.54 ± 0.59 ^a^
n-Butanol	100.86 ± 0.63 ^b^
Water	5.92 ± 0.30 ^f^

^1^ GAE: gallic acid equivalents; ^2^ Values are mean ± SD (n = 3); ^3^ Values with different letters within the same column (a–f) differ significantly (*p* < 0.05).

**Table 3 plants-13-03123-t003:** FRAP values of various solvent fractions from methanol extract of *Cynanchum thesioides* (Freyn) K. Schum stems, leaves and buds.

Extraction Solvent	FRAP (M/g)
Methanol	1.52 ± 0.01 ^1d2^
n-Hexane	1.20 ± 0.00 ^e^
Dichloromethane	2.32 ± 0.01 ^b^
Ethyl acetate	3.49 ± 0.00 ^a^
n-Butanol	2.24 ± 0.01 ^c^
Water	0.93 ± 0.00 ^f^

^1^ Values are mean ± SD (n = 3); ^2^ Values with different letters within the same column (a–f) differ significantly (*p* < 0.05).

**Table 4 plants-13-03123-t004:** DPPH radical scavenging activity of various solvent fractions from methanol extract of *Cynanchum thesioides* (Freyn) K. Schum stems, leaves and buds.

Extraction Solvent	DPPH Radical Scavenging Activity (IC_50_, mg/mL) ^1^
Methanol	1.22 ± 0.00 ^2d3^
n-Hexane	6.12 ± 0.41 ^f^
Dichloromethane	0.73 ± 0.00 ^c^
Ethyl acetate	0.42 ± 0.00 ^b^
n-Butanol	0.48 ± 0.01 ^b^
Water	2.24 ± 0.03 ^e^
Ascorbic acid	0.01 ± 0.00 ^a^

^1^ Inhibitory activity was expressed as the mean of 50% inhibitory concentration determined in triplicate, obtained by interpolation of concentration inhibition curve; ^2^ Values are mean ± SD (n = 3); ^3^ Values with different letters within the same column (a–f) differ significantly (*p* < 0.05).

**Table 5 plants-13-03123-t005:** ABTS radical scavenging activity of various solvent fractions from methanol extract of *Cynanchum thesioides* (Freyn) K. Schum stems, leaves and buds.

Extraction Solvent	ABTS Radical Scavenging Activity (IC_50_, mg/mL) ^1^
Methanol	1.61 ± 0.00 ^2e3^
n-Hexane	9.45 ± 0.02 ^g^
Dichloromethane	0.92 ± 0.01 ^d^
Ethyl acetate	0.43 ± 0.01 ^b^
n-Butanol	0.60 ± 0.00 ^c^
Water	3.48 ± 0.01 ^f^
Ascorbic acid	0.07 ± 0.00 ^a^

^1^ Inhibitory activity was expressed as the mean of 50% inhibitory concentration determined in triplicate, obtained by interpolation of concentration inhibition curve; ^2^ Values are mean ± SD (n = 3); ^3^ Values with different letters within the same column (a–g) differ significantly (*p* < 0.05).

**Table 6 plants-13-03123-t006:** Reducing power of various solvent fractions from methanol extract of *Cynanchum thesioides* (Freyn) K. Schum stems, leaves and buds.

Extraction Solvent	Reducing Power (EC_50_, mg/mL) ^1^
Methanol	1.40 ± 0.01 ^2c3^
n-Hexane	10.36 ± 0.30 ^e^
Dichloromethane	1.38 ± 0.03 ^c^
Ethyl acetate	0.87 ± 0.00 ^b^
n-Butanol	0.95 ± 0.00 ^b^
Water	1.92 ± 0.00 ^d^
Ascorbic acid	0.04 ± 0.00 ^a^

^1^ Effective activity was expressed as the mean of 50% effective concentration determined in triplicate, obtained by interpolation of concentration inhibition curve; ^2^ Values are mean ± SD (n = 3); ^3^ Values with different letters within the same column (a–e) differ significantly (*p* < 0.05).

**Table 7 plants-13-03123-t007:** Superoxide dismutase (SOD)-like activity of various solvent fractions from methanol extract of *Cynanchum thesioides* (Freyn) K. Schum stems, leaves and buds.

Extraction Solvent	SOD-Like Activity (%)
Methanol	14.72 ± 1.36 ^1d2^
n-Hexane	18.92 ± 0.41 ^c^
Dichloromethane	28.17 ± 1.12 ^b^
Ethyl acetate	30.36 ± 0.44 ^a^
n-Butanol	28.98 ± 1.52 ^ab^
Water	19.20 ± 0.93 ^c^

^1^ Values are mean±SD (n = 3); ^2^ Values with different letters within the same column (a–d) differ significantly (*p* < 0.05).

## Data Availability

Data are contained within the article.

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
