# Peer review of "Study on Bioactive Components of Aromatic Cynanchum thesioides (Freyn) K. Schum by Solvent Fractionation"

_plants, 2024, doi:10.3390/plants13223123_

Round 1
Reviewer 1 Report
Comments and Suggestions for Authors
General comments
· In overall, I found the work very interesting and in line with the Journal scope. Presentation and writing style is good with some lapses in style.
· Title: correct “bioactive”
· Abstract: it well written and main data results are presented.
· Methodology: Good description and coherent.
· Analysis and results: Data well-presented. However many correlations could be presented with the analysis data
· Conclusions: total phenol content was not underlined.
· References: Solely c.a. 42% οf References are up-to-date.
Specific comments (list not exhaustive)
Introduction:
Are there other studies on this specific herb?Limited but they are not undelined.
Results:
L99: which the equivalent for polyphenol content
L104: Did you perform HPLC analysis? In order to state the phenolic cmpounds?
L117: please underline the Reference
L138-139: Could you investigate any correlation between phenols and antioxidant and/or/ scavenging activity?
L168: How could the authors explain the increased ABTS activity when using water?
Materila & Methods:
How many samples did the authors analyse?
Conclusions:
L338: Did the authors consider the cost of such a solvent?

Presentation and writing style is good with some lapses in style.
Author Response
Dear Reviewer, thank you for taking the time out of your busy schedule to review our paper and for offering valuable revision suggestions. We have carefully considered each of your specific comments and revised them accordingly.
General comments
-In overall, I found the work very interesting and in line with the Journal scope. Presentation and writing style is good with some lapses in style.
Answer: Thank you. As you mentioned, we will further refine specific details in the manuscript to enhance clarity and professionalism, aligning it closely with the journal's standards.
- Title: correct “bioactive”.
Answer: Thank you. As you mentioned, the title was changed to “Study on Bioactive Components of Aromatic Cynanchum thesioides (Freyn) K. Schum by Solvent Fractionation (Page 1, line 2~3).
- Analysis and results: Data well-presented. However many correlations could be presented with the analysis data.
Answer: Thank you. As you mentioned, in the revised manuscript, we have incorporated certain relevant correlation analyses in the Results and Discussion section to strengthen data interpretation (Page 2~6, line 92~221).
- Conclusions: total phenol content was not underlined.
Answer: Thank you. As you mentioned, in the revised manuscript, we have further emphasized the role of total phenol content in influencing antioxidant activities, particularly highlighting the ethyl acetate fraction, which exhibited the highest polyphenol content and corresponding antioxidant activity (Page 3, line 102~113; Page 10, line 350~367).
- References: Solely c.a. 42% οf References are up-to-date.
Answer: Thank you. As you mentioned, we have reviewed the reference list and added recent literature from recent years (pp. 10-11, lines 380-451).
Specific comments (list not exhaustive)
- Introduction:
Are there other studies on this specific herb?Limited but they are not undelined.
Answer: Thank you. As you mentioned, in the revised manuscript, we have expanded the Introduction to include additional studies on Cynanchum thesioides, emphasizing the significance and novelty of the current research (Page 2, line 56~73).
- Results:
L99: which the equivalent for polyphenol content
Answer: Thank you. As you mentioned, in the revised manuscript, we have added the unit "gallic acid equivalents (GAE)" in the 2.2 Total Polyphenol Content section to clearly indicate the basis for polyphenol content measurements. (Page 3, line 102).
L104: Did you perform HPLC analysis? In order to state the phenolic cmpounds?
Answer: Thank you for the suggestion. Unfortunately, the HPLC equipment at our institution was temporarily unavailable due to technical issues during the study. However, we plan to incorporate HPLC analysis in future studies to further characterize the phenolic compounds.
L117: please underline the Reference
Answer: Thank you. As you mentioned, we have ensured the reference is clearly cited in the revised manuscript. (Page 10~11, line 380~451).
L138-139: Could you investigate any correlation between phenols and antioxidant and/or/ scavenging activity?
Answer: Thank you for your suggestion regarding the investigation of correlations between phenols and antioxidants and/or/ scavenging activity. We have added discussions in the DPPH and ABTS results sections to highlight the significant role of total phenol content in contributing to the observed antioxidant activities. This addition enhances the clarity and depth of our findings (Page 4, line 151~158; Page 5, line 176~192).
L168: How could the authors explain the increased ABTS activity when using water?
Answer: Thank you for your insightful question regarding the increased ABTS activity observed when using the water fraction. One possible explanation is that the water fraction may contain a variety of hydrophilic antioxidant compounds, including certain phenolic acids and flavonoids, which can effectively scavenge ABTS radicals. Additionally, the extraction conditions may favor the solubility of these compounds in the water fraction, enhancing their overall antioxidant capacity. To further investigate this phenomenon, we plan to incorporate analytical methods such as HPLC and GC in our future experiments to identify and quantify the specific antioxidant compounds present in the water fraction. This will allow us to gain a deeper understanding of the underlying mechanisms contributing to the observed ABTS activity. We will clarify this point in the discussion section of the manuscript. (Page 5, line 184~192).
- Materila & Methods:
How many samples did the authors analyse?
Answer: Thank you for your inquiry regarding the number of samples analyzed in our study. For the extraction and fractionation of Cynanchum thesioides, we used six different solvent fractions: Methanol, n-Hexane, Dichloromethane, Ethyl acetate, n-Butanol, and Water. In our initial experiment, we performed three extractions for each solvent. Subsequently, for each fraction, we conducted seven different antioxidant assays. That for each of the six solvent fractions, we analyzed a total of 21 assays (3 extractions × 7 assays). To ensure the reliability of our results, we repeated this entire experimental process two additional times, resulting in a total of three independent experimental runs. The average data from these three experiments were used for our analysis.
- Conclusions:
L338: Did the authors consider the cost of such a solvent?
Answer: Thank you for your valuable feedback. In our manuscript, we have addressed the economic aspects of solvent selection, particularly in relation to the ethyl acetate and n-butanol fractions. Our findings indicate that n-butanol, with a yield of 5.80%, significantly surpasses the ethyl acetate fraction’s yield of 1.29%, making it a more economically viable option. We recognize the importance of these considerations for future product development and commercialization. In the revised manuscript, we have expanded our discussion on economic evaluations to further substantiate our conclusions (Page 10, line 350~369).
Thank you for reviewing our manuscript. If you have any questions during the review, please feel free to contact the author at any time.

Reviewer 2 Report
Comments and Suggestions for Authors
I carefully read the manuscript entitled “Study on Bioactivie Components of Aromatic Cynanchum thesioides (Freyn) K. Schum by Solvent Fractionation”. Such a topic is intriguing in light of the search for natural bioactive components. Overall, I think the article contributes a considerable amount of valuable knowledge. However, there are some aspects that need further clarification.
1. The authors mention economics. In this kind of research, it is important to present the main assumptions that prove in favour of conducting and implementing such projects.
2. In several places there are potential errors. Please check and verify.
Line 29: is “aging” and should be “ageing”
Line 32: is “defense” and should be “defense”
Line 52 is “diarrhea” and should be “diarrhoea”
Lines 58 and 215 are “anti-aging” and should be “anti-ageing”
Lines 124, 151, 173, 197 and 220-221 are “letter within a some” and should be “letters within the some”
Lines 124-125, 151, 173, 197 and 221 are “significant” and should be “significantly”
Lines 149-150 is “determines” and should be “determined”
Line 155 is “decolorized” and should be “decolored”
3. What type of statistical evaluation did the authors employ, given that they presented a p-value of less than 0.05 with the following results? This should be completed.
Overall, I find the article intriguing and logically composed. The purpose of my comment is not to diminish the overall quality of this manuscript but to provide the scientific community with a better understanding of the issue highlighted. I believe that this manuscript will arouse the interest of readers. With the indicated corrections, I strongly recommend the manuscript for publication.
Author Response
Dear Reviewer, thank you for taking the time out of your busy schedule to review our paper and for offering valuable revision suggestions. We have carefully considered each of your specific comments and revised them accordingly.
Comments and Suggestions for Authors
1. The authors mention economics. In this kind of research, it is important to present the main assumptions that prove in favour of conducting and implementing such projects.
Answer:
Thank you for your insightful comment regarding the economic considerations of our research. We acknowledge the importance of presenting the main assumptions that support the feasibility of conducting and implementing such projects. In our study, we recognize that the economic viability of utilizing Cynanchum thesioides and its solvent fractions for product development is crucial. The following assumptions underline the economic rationale behind our research:
1.1. Market Demand: There is a growing interest in natural antioxidants derived from plant sources due to their potential health benefits. This trend suggests a strong market for products derived from Cynanchum thesioides, particularly in the food, cosmetics, and pharmaceutical industries.
1.2. Cost-Effectiveness of Extraction: The extraction process using n-butanol, which yielded the second-highest antioxidant activity at a more favorable yield, indicates a cost-effective approach compared to other solvents. This supports the economic feasibility of using n-butanol for large-scale production.
1.3. Sustainability: The use of a natural herb like Cynanchum thesioides aligns with sustainable development goals, as it promotes the utilization of renewable resources while minimizing environmental impact.
1.4. Potential for Commercialization: The antioxidant properties of the extracts, coupled with their potential applications, create opportunities for commercialization. The identified solvent fractions can be further optimized for specific products, thus enhancing their marketability.
2. In several places there are potential errors. Please check and verify.
2.1. Line 29: is “aging” and should be “ageing”
Answer: Thank you. As you mentioned, we appreciate your attention to detail regarding the spelling of "aging." We have corrected "aging" to "ageing" in the revised manuscript (Page 1, line 29).
2.2. Line 32: is “defense” and should be “defense”
Answer: Thank you. As you mentioned, we appreciate your attention to detail regarding the spelling of "defense." We have corrected "defense" to "defence" in the revised manuscript (Page 1, line 32).
2.3. Line 52 is “diarrhea” and should be “diarrhoea”
Answer: Thank you. As you mentioned, we appreciate your attention to detail regarding the spelling of "diarrhea." We have corrected "diarrhea" to "diarrhoea" in the revised manuscript (Page 2, line 52).
2.4. Lines 58 and 215 are “anti-aging” and should be “anti-ageing”
Answer: Thank you. As you mentioned, we appreciate your attention to detail regarding the spelling of "anti-aging." We have corrected "anti-aging" to "anti-ageing" in the revised manuscript (Page 2, line 60; Page 6, line 239).
2.5 Lines 124, 151, 173, 197 and 220-221 are “letter within a some” and should be “letters within the some”
Answer: Thank you. As you mentioned, we appreciate your attention to detail regarding the sentences of "letter within a some." We have corrected "letter within a some" to "letters within the some" in the revised manuscript (Page 3, line 116; Page 4, line 133; Page 5, line 163; Page 5, line 197; Page 6, line 221; Page 7, line 244).
2.6 Lines 124-125, 151, 173, 197 and 221 are “significant” and should be “significantly”
Answer: Thank you. As you mentioned, we appreciate your attention to detail regarding the sentences of "significant" We have corrected "significant" to "significantly" in the revised manuscript (Page 3, line 117; Page 4, line 134; Page 5, line 163; Page 5, line 197; Page 6, line 221; Page 7, line 245).
2.7 Lines 149-150 is “determines” and should be “determined”
Answer: Thank you. As you mentioned, we appreciate your attention to detail regarding the spelling of "determines" We have corrected "determines" to "determined" in the revised manuscript (Page 5, line 162; Page 5, line 196; Page 6, line 219).
2.8 Line 155 is “decolorized” and should be “decolored”
Answer: Thank you. As you mentioned, we appreciate your attention to detail regarding the spelling of "decolorized" We have corrected "decolorized" to "decolored" in the revised manuscript (Page 5, line 167).
3. What type of statistical evaluation did the authors employ, given that they presented a p-value of less than 0.05 with the following results? This should be completed.
Answer: Thank you for your valuable comment regarding the statistical evaluation. We employed one-way analysis of variance (ANOVA) followed by Tukey’s post-hoc test to analyze the differences between groups, with significance set at p < 0.05. In the revised manuscript, we will clarify the statistical methods used to ensure the results are appropriately supported by our analysis.
Thank you for reviewing our manuscript. If you have any questions during the review, please feel free to contact the author at any time.
